# MicroRNAs as Candidate Biomarkers for Alzheimer’s Disease

**DOI:** 10.3390/ncrna7010008

**Published:** 2021-02-01

**Authors:** Colin Kanach, Jan K. Blusztajn, Andre Fischer, Ivana Delalle

**Affiliations:** 1Department of Pathology and Laboratory Medicine, Rhode Island Hospital, Lifespan Academic Medical Center, The Warren Alpert Medical School of Brown University, Providence, RI 02903, USA; ckanach@lifespan.org; 2Department of Pathology and Laboratory Medicine, Boston University School of Medicine, 670 Albany Street, Boston, MA 02118, USA; jbluszta@bu.edu; 3German Center for Neurodegenerative Diseases, Department for Epigenetics and Systems Medicine in Neurodegenerative Diseases, 37075 Göttingen, Germany; a.fischer@eni-g.de

**Keywords:** plasma microRNAome, MCI, epigenetic dysregulation

## Abstract

The neurological damage of Alzheimer’s disease (AD) is thought to be irreversible upon onset of dementia-like symptoms, as it takes years to decades for occult pathologic changes to become symptomatic. It is thus necessary to identify individuals at risk for the development of the disease before symptoms manifest in order to provide early intervention. Surrogate markers are critical for early disease detection, stratification of patients in clinical trials, prediction of disease progression, evaluation of response to treatment, and also insight into pathomechanisms. Here, we review the evidence for a number of microRNAs that may serve as biomarkers with possible mechanistic insights into the AD pathophysiologic processes, years before the clinical manifestation of the disease.

## 1. Introduction

Neurodegenerative diseases are chronic illnesses that cause progressive and protracted neurological decline, with late onset, sporadic Alzheimer’s disease (AD) being the most common. AD results in devastating cognitive, emotional, physical, social, and financial consequences for the patients and their families alike. Impaired cognitive function is a hallmark of AD and often one of the first symptoms. However, changes in cognitive function develop slowly over time, and patients are diagnosed at an advanced stage of cellular and molecular pathology [1,2]. The mechanisms that initiate AD and enable its progression are not well understood but result in the unrelenting spread of neuronal damage, including loss of synapses, metabolic and structural abnormalities, inflammation, and ultimately cell death, believed to be strongly related to the combination of pathohistological hallmarks, neuritic plaques and neurofibrillary tangles, and their building blocks, amyloid-beta and tau, respectively [3]. Extracellular amyloid-beta (Aβ40–42) peptide deposits in the form of neuritic plaques are the *sine qua non* for AD diagnosis on brain autopsy; however, the degree of amyloidosis does not correlate with the severity of AD symptoms [4]. Intracellular hyperphosphorylated tau accumulations give rise to the second AD hallmark, neurofibrillary tangles, but the biochemical basis for aberrant tau phosphorylation in AD is poorly understood [5]. Nevertheless, cerebrospinal fluid (CSF) levels of Aβ42, t-tau (total tau), and p-tau (tau phosphorylated at threonine 181) are considered core biomarkers to support AD diagnosis [6,7,8]. Two major problems are associated with current, conventional AD biomarkers.

First, as recognized by the National Institute of Aging and Alzheimer’s Association (NIA-AA), the current biomarkers are relatively difficult to access because the mechanisms that isolate and protect the central nervous system from injury also act as barriers to diagnosis [9]. To that end, recently reported plasma biomarker p-tau217 represents a step forward and a promise as p-tau217 plasma levels distinguished AD from other neurodegenerative diseases in multiple international cohorts [10]. However, it is unknown how early in the trajectory of AD p-tau217 plasma levels start to increase, which represents the remaining gap of knowledge about prognostic AD biomarkers that could identify the population at risk and the population in which therapeutic interventions may be deployed to halt molecular events predating cognitive decline. Second, cortical measurements of amyloid-beta and tau depositions via magnetic resonance imaging (MRI) and positron emission tomography (PET) [9] are expensive and thus not accessible on the scale needed for a public health challenge such as AD [8,11]. Last but not the least, the NIA-AA research framework aims to define AD biologically and thus seeks to expand AD biomarkers according to the scientific progress in the field of AD pathogenesis [9].

## 2. Small Non-Coding RNAs Regulate Brain Health

The failure to diagnose AD at an early stage of molecular pathology is considered to be the major reason why treatments have failed in clinical trials [12]. The hippocampus is among the first brain structures to show AD-associated pathology—neuritic plaques, neurofibrillary tangles, and synaptic and neuronal loss [3,13,14]. Deregulated hippocampal gene expression has been observed during aging and neurodegeneration in animal models and in human patients [15,16,17,18,19,20]. Small non-coding RNAs (ncRNAs) regulate gene expression in response to genome–environment interactions. In order to be applicable in routine screenings in primary care settings, AD biomarkers must be minimally invasive and inexpensive while able to identify individuals in need of further, more invasive, and more expensive diagnostic procedures [21]. Circulating small ncRNAs are expected to serve as such biomarkers [22,23,24,25,26,27,28,29,30,31,32]. The purpose of this review is to highlight evidence that evaluating circulating levels of a set of small ncRNAs might serve as AD prognostic and diagnostic biomarkers for consideration by the NIAA-AA research framework [9]. We deem this consideration to be particularly timely because RNA therapeutics are emerging as a promising approach in central nervous system (CNS) diseases [33,34,35,36]. 

The vast majority (~95%) of circulating small ncRNAs are microRNAs (miRNAs, miRs), ~19–22 nucleotides long, evolutionary conserved, and most extensively studied members of the ncRNA family [37]. They are transcribed by RNA polymerase II or RNA polymerase III into primary miRNA (Pri-miRNA) and further digested by microprocessor complexes to a Pre-miRNA, which is a hairpin loop-like structure roughly 70 nucleotides in length. This structure is then transported out of the nucleus into the cytoplasm via an exportin five transporter and then is processed by Dicer and TAR-RNA binding proteins (TRBP) to a miRNA duplex structure. Helicase further separates the duplex structure into two single strands, a guide strand and a passenger strand. The passenger strand is degraded, while the guide strand goes on to complex with Ago2 proteins to form the mature miRNA and serves as a regulator of post-transcription gene expression via pairing with messenger RNA (mRNA) [37]. Accumulated evidence suggests that miRNAs are secreted in extracellular space in vesicle-free form or in a microvesicle-encapsulated form bound to Ago2 proteins that provide resistance to nucleases, conveying stability to miRNAs in biofluids [38]. Consistently, miRNAs are extremely stable in cell-free environments and resistant to thaw–freeze cycles [23].

Although aging represents the largest risk for sporadic, late onset AD, sociological, medical, and nutritional factors are thought to play a role [39,40]. Some of these epigenetic factors become influential rather early in life and under the umbrella term of “early life stress (ELS)” affect the onset of dementia as shown in monozygotic twin studies [41,42]. ELS rodent models suggest specific pathways through which genomic and epigenomic interaction affects neuroinflammation, cerebral lipid metabolism, brain insulin resistance, and myelin disintegration, in a manner that may ultimately lead to AD-associated histopathology [39].

A recent review of 191 miRNAs across 35 human studies measuring miRNA levels in blood, serum or plasma identified 30 miRNAs, replicated in more than one study, as dysregulated miRNAs in psychoses [43]. These 30 miRNAs are known to regulate pathways governing brain development and plasticity [43]. Consistently, depletion of the enzyme needed for miRNA biogenesis, Dicer, results is neurodevelopmental abnormalities including reduced progenitor numbers, abnormal neuronal differentiation, and thinner cortical wall in the mouse brain [44]. Further, miRNAs expressed in the human brain [45] regulate neuronal differentiation and synaptic plasticity [46,47]. Recent studies point toward a widespread involvement of ncRNA families in local translation within neuronal cells, including less abundant small RNAs (PIWI-interacting RNAs (piRNAs), endogenous small interfering RNAs (endo-siRNAs)) and long ncRNAs (circular RNAs (circRNAs), long intergenic ncRNAs (lincRNAs)). The emerging picture is an intricate crosstalk between different ncRNA families, mRNAs, and RNA-binding proteins (RBPs) orchestrating local dendritic proteome in an activity-dependent manner. The alterations in the activity of dendritic miRNAs, miR-132, miR-134, and miR-138 at the synapse have been linked with the pathogenesis of psychoses and neurodegenerative diseases [48], as the role of miRNAs in learning and memory in health and in AD has been extensively documented [33,49,50,51,52,53]. Consistently, miRNAs implicated in cognitive dysfunction associated with neurodegeneration were identified in CSF and postmortem brains of AD patients and also in mouse models of AD [33,54].

## 3. miRNAs as Diagnostic and Prognostic AD Biomarker Candidates

The majority of studies focused on epigenetic dysregulation as the source of biomarkers of preclinical AD have focused on the use of peripheral blood. Peripheral blood mononuclear cells (PBMCs) are one of the major cellular components to blood and are one the major sources of miRNAs [23]. They are of particular interest due to their innate ability to respond to internal and external stimuli and store information via epigenetic mechanism including DNA methylation, histone acetylation, and small ncRNA, including miRNA [23]. As potential biomarkers for prodromal and presymptomatic stages of AD, miRNAs, thanks to novel methods for their detection, represent a cost-efficient approach to screening persons for AD risk to identify individuals in need for further evaluation such as CSF analysis and imaging studies measuring AT(N) markers. 

Kenny et al. (2019) evaluated diagnostic and prognostic potential of plasma miRNA profiling by seeking to differentiate between aged controls and patients with MCI and AD and also to predict cognitive decline in a five-year longitudinal study [55]. The discovery cohort included 31 control subjects and 30 patients with minimal cognitive impairment (MCI) enrolled in single-center community-based study with neurological, neuropsychological, and genetic information available. The longitudinal cohort included 18 subjects with progressions from cognitively healthy to dementia or stable MCI with longitudinal samples collected at 1–2-year intervals over a 5-year time period. miR-206 emerged as a biomarker for AD and for predicting the conversion from MCI to AD [55]. Altered levels of miR-206 in MCI and AD may be related to its target, brain-derived neurotrophic factor (BDNF) [56]. BDNF protein and mRNA levels have been reported to decrease in individuals with MCI and AD, resulting in synaptic loss and cell death, central to AD pathology [57,58,59,60].

Two recent studies, through diverse approaches, assessed performance of CSF miRNAs as biomarkers for cognitive decline. Using next-generation sequencing to study small ncRNAome in CSF exosomes, Jain and colleagues (2019) identified a combined signature consisting of three miRNAs and three piRNAs suitable to detect AD with an area under the curve (AUC) of 0.83 in a replication cohort and, importantly, predicted the conversion of MCI patients to AD with an AUC of 0.86 for the piRNA signature [54]. When combined with pTau and Aβ 42/40 ratio, the small ncRNA signature yielded an AUC of 0.98 [54]. Sandau and colleagues (2020) measured the expression of 17 miRNAs, implicated in AD based on available literature, in CSF samples from AD, MCI, and cognitively intact controls with the objective to test the ability of these implicated miRs to identify prodromal AD in MCI [61]. Five miRNAs (miR-142-3p, miR-146a-5p, miR-146b-5p, miR-193a-5p, miR-365a-3p) showed a linear trend of decreasing median expression across the ordered diagnoses (control to MCI to AD) and, importantly, when combined with Aβ42:T-tau levels, predicted MCI with increased classification performance [61]. The five CSF miRNAs suggested to predict AD in subjects with MCI [61] did not overlap with three CSF exosomal miRNAs (miR-34c, miR-30a, and miR-27a) within the small ncRNA signature to detect and predict conversion to AD [54], but common to both sets were the predicted targets-various cellular processes maintaining neuronal homeostasis, including response to pathologic stress [62].

Here, we review the evidence from multiple studies in support of the notion that a few miRNAs may serve as prognostic and diagnostic AD biomarkers with a mechanistic dimension. This group of candidate AD biomarker miRNAs highlights dysregulation in cellular functions underlying pathophysiologic processes likely years before the manifestation of AD symptoms. 

### 3.1. miR-29

β-site amyloid precursor protein cleaving enzyme 1 (BACE1) is a proteolytic enzyme that cleaves full-length amyloid precursor proteins (APP), contributing to the accumulation of neuritic plaques containing β-amyloid peptide Aβ40–42 in the neuropil, the AD hallmark. Hebert et al. (2008) reported high BACE 1 expression in postmortem brains of AD patients as well as a decreased expression of several miRNAs with candidate binding sites within 3′ UTRs of *BACE1* [51]. They went on to demonstrate that BACE1 is a miRNA target gene and that the levels of expression of BACE1 and miR-29 a/b-1 do correlate in AD brains. The analysis of CSF samples of AD patients and non-demented controls by Muller et al. (2016) confirmed the association between the level of miR-29a expression and AD diagnosis [63]. miR-29a-3p levels have been reported as decreased in CSF of AD patients in other studies as well, although not always consistently [61]. To demonstrate the mechanistic relationship, Hebert et al. performed a series of in vitro functional essays proving that miR-29a/b-1 indeed can modulate BACE1 activity and thus β-amyloid production [51]. Excitingly, Pereira and colleagues (2016) developed technology to successfully deliver the recombinant pre-miR-29b into the cytoplasm in vitro, eliciting decreased expression of BACE1 and Aβ42 [64]. Together, these results suggest the pathophysiological, disease-mechanism-based connection between a level of a miRNA expression in clinically accessible biospecimens and the levels of an enzyme responsible for the accumulation of a histological hallmark of the disease. Equally exciting are the promising inroads into technology needed for successful delivery of miRNA-based therapeutics [64].

### 3.2. miR-34

A major risk factor for AD is aging, a process with many regulating genes including *SIRT* genes that encode sirtuins, proteins that govern cellular functions negatively impacted by aging including metabolism and response to oxidative stress and to DNA damage. Owczarz and colleagues (2017) showed that the PBMCs of long-lived individuals contained a low expression of *SIRT1* mRNA coupled with high expression of miR-34a [65]. Data mining techniques to identify miRNA that both have a large number of targets and are differentially expressed in brains of AD patients identified increased expression of miR-34a in the temporal cortex of AD patients to correlate with the severity of AD pathology and also reported increased expression of miR-34a in 3xTg-AD mouse model [66]. Prior to those studies, Zovoilis and colleagues (2011) found miR-34c to be upregulated in the hippocampus of mouse models for amyloid deposition and age-associated memory decline and also in brain tissue from AD patients [33,67]. This group went further to show that the inhibition of miR-34c function ameliorated memory impairment and gene expression in a mouse model for amyloid deposition [33]. miR-34c expression is thought to grow in response to pathological stress [62] consistent with the evidence that stress-induced gene deregulation promotes cognitive decline [68]. Remarkably, miR-34c also belonged to a 6-member small ncRNA signature obtained from CSF exosomes suitable to detect AD and predict the conversion of MCI patients to AD [54]. The mechanistic dimension of miR-34 expression level as an AD biomarker is further strengthened by the evidence for the role of miR-34 in aging and neurodegeneration in *Drosophila* [69] and *Caenorhabditis elegans* [70], highlighting the evolutionary conserved regulation of fundamental cellular functions by miRNAs.

### 3.3. miR-146

In addition to miR34a, miR146a was in the 5-member group of miRNAs studied in the context of AD based on the evidence from data mining from various miRNA target gene databases predicting their role in cellular functions thought to be compromised in AD [66]. Furthermore, in the analysis of CSF samples from AD, MCI, and cognitively normal controls, miR-146a-5p was among five CSF miRNA with levels decreasing as normal controls developed MCI and MCI progressed to AD [61]. miR-146a-5p was also among four miRNAs that were upregulated in mouse models of AD (APPtg and TAUtg) and also in the brains of AD patients [71]. Interestingly, induction of miR-146a via gastrointestinal Gram-negative bacteria is thought to promote pathogenic stimulation of innate–immune and neuroinflammatory pathways relevant for inflammation-mediated pathologic consequences of amyloidosis in AD brain [72].

Additional miRNAs reported to be differentially expressed in postmortem brain tissue, CSF, PBMC, plasma, or serum of AD patients are presented in Table 1.

## 4. Conclusions

The accumulated evidence suggests that harnessing the power of ongoing large epidemiologic studies could provide robust classifier enabling the correlation of cognitive status in densely phenotyped individuals with plasma miRNAome to create and expand the tools for predicting and diagnosing AD. As shown in Diagram 1, it would then be possible to define the relationship between the expression of a circulating miRNA biomarker and the expression of that biomarker in the brain cells central to AD pathophysiology. Finally, it would be feasible to understand the functional consequence of a human disease miRNA biomarker in models of AD to enable mechanistic approaches toward the design of novel preventive and therapeutic interventions (Figure 1). 

## Figures and Tables

**Figure 1 ncrna-07-00008-f001:**
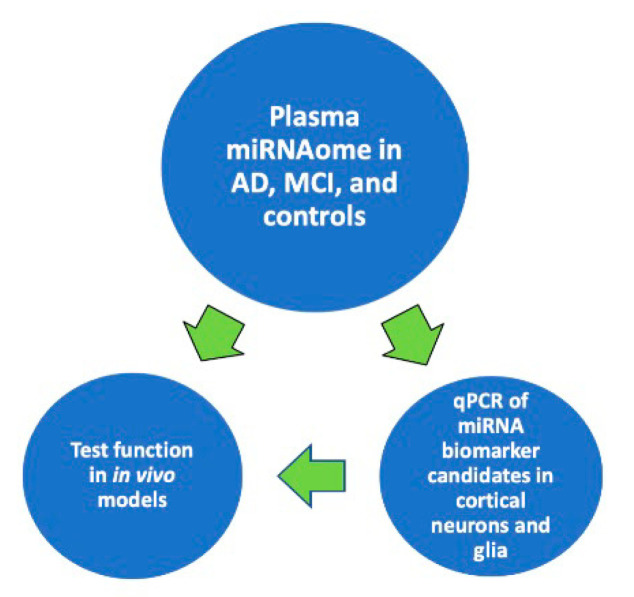
Proposed approach toward discovering and evaluating circulating miRNA biomarker candidates for prognosing and diagnosing AD-associated cognitive decline.

**Table 1 ncrna-07-00008-t001:** Additional miRNAs reported to be differentially expressed in postmortem brain tissue, cerebrospinal fluid (CSF), peripheral blood mononuclear cells (PBMCs), plasma, or serum of Alzheimer’s disease (AD) patients.

miRNA	Subjects	Material	Results	Reference
miR-125b	10 AD patients, 5 controls	frontal cortex	increased in AD patients, correlating with increased tau phosphorylation	Banzhaf-Strathmann, Benito et al., 2014 [34]
miR-132-3p	hippocampi from 41 AD patients and 23 controls; prefrontal cortices from 49 individuals at Braak stage I-VI and 7 controls	hippocampus, prefrontal cortex	decreased in AD	Lau, Bossers et al., 2013 [73]
miR-107	19 individuals, Braak stage 0-VI	temporal cortex (Brodmann area 21/22)	decreasing with increased Braak stage	Nelson and Wang, 2010 [74]
miR-212 miR-132	brains from 11 AD, 7 “high pathology controls”, 9 controls; plasma from 16 AD patients, 16 AD-MCI subjects, and 31 controls	brains and neurally derived plasma exosomes	decreased in AD brains and neurally-derived plasma exosomes	Cha, Mengel et al., 2019 [75]
12 miR signature	48 AD patients, 22 controls	PBMC	separated AD patients from controls with 93.3% accuracy	Leidinger and Backes et al., 2013 [26]
miR-146a	30 AD patients, 28 controls	Plasma	no correlation with AD diagnosis, but correlation with age and lower MMSE scores	Maffioletti, Milanesi et al., 2020 [76]
miR-501-3p	27 AD patients, 18 controls	Serum	decreased in AD compared to age-matched controls; lower expression correlated with lower MMSE scores	Hara, Kikuchi et al., 2017 [30]
miR-125b	22 AD patients, 18 subjects with non-inflammatory, and 8 subjects with inflammatory neurological conditions; 10 frontotemporal dementia patients	Serum	decreased in AD compared to non-inflammatory neurological conditions	Galimberti and Villa et al., 2014 [27]

## Data Availability

No new data were created or analyzed in this study. Data sharing is not applicable to this article.

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
