# Peer review of "MicroRNAs as Candidate Biomarkers for Alzheimer’s Disease"

_ncrna, 2021, doi:10.3390/ncrna7010008_

Round 1
Reviewer 1 Report
This review addresses an important and topical issue: the use of miRNAs as potential biomarkers of AD. Given the paucity of biomarkers of AD, especially in early, pre-clinical stages, the identification of miRNAs that may correlate with AD would be of high clinical importance. In recent years miRNAs have emerged as a versatile new class of gene regulatory elements of relevance in human disease. This well-written review gives a good general overview that describes the roles of miRNAs in AD that would be of general interest to the audience of this journal. However, the authors could strengthen their review by more clearly emphasizing the conservation of miRNAs across phyla and supportive evidence for miRNA function in other animal in vivo AD models. Some suggestions for improvements and other comments follow below.
General comment: Although this article gives a good overview of miRNA - including their biogenesis and function, it fails to highlight the role of animal models in our understanding of their function in neurobiology and in aging-associated diseases. Indeed, many pioneering studies (including the discovery of miRNAs) were done in these animal models and it would be worthwhile to emphasize how conserved many of the miRNAs area across metazoans – thus lending further credence to the argument that investigating miRNAs as biomarkers of AD is worthwhile and scientifically reasonable. Some specific examples follow below.
Specific comments:
l.64-66 – Although several good reviews are mentioned another good review to cite on the topic would be Basavaraju et al, 2016 – as it highlights many of the same miRNAs that will be discussed in this review.
l.88 – This paragraph connects epigenetic factors associated with AD (previous paragraph) to small RNAs as epigenetic factors of neurobiological relevance. Although several interesting examples are mentioned it would be worthwhile to also mention pioneering examples of small RNAs associated with aging-associated neurodegeneration in model organisms such as Drosophila (i.e. miR-34 from Liu N, Nature 2012) – particularly as miR-34 will be a focus of interest in this review.
l.82, 87, 93: minor. Space between text and reference is inconsistent in this section (and perhaps others).
l.171 – 178. The section on miR-34 would benefit from reference to other studies such as Liu N, Nature 2012 and Yang J, Age, 2013 that indicate aging-associated functions for miR-34 in other animal models.
l.189: “a trend of decreased progression” is vague. Could be re-written for clarity.
Diagram 1. Why limit only to mouse models? Recommended editing 3rd bubble to "Test miRNA function in AD in vivo models."
Author Response
We appreciate the constructive and helpful critiques that we were pleased to consider. We have addressed all the specific comments in the revised manuscript as follows:
l.64-66 – "Although several good reviews are mentioned another good review to cite on the topic would be Basavaraju et al, 2016 – as it highlights many of the same miRNAs that will be discussed in this review." We have included this reference (l.66 of the revised manuscript, Ref# 32).
l.88 – "...it would be worthwhile to also mention pioneering examples of small RNAs associated with aging-associated neurodegeneration in model organisms such as Drosophila (i.e. miR-34 from Liu N, Nature 2012) – particularly as miR-34 will be a focus of interest in this review." AND "l.171 – 178. The section on miR-34 would benefit from reference to other studies such as Liu N, Nature 2012 and Yang J, Age, 2013 that indicate aging-associated functions for miR-34 in other animal models." We thank the reviewer for these important references and also for the general comment regarding the inclusion of studies that highlight evolutionary conserved roles of microRNAs. They are both included (Refs #69 and #70) and a highlighted text is added in l. 188-191 of the revised manuscript.
l.189: "“a trend of decreased progression” is vague. Could be re-written for clarity". We have done so in the highlighted l. 197 of the revised manuscript.
Diagram 1. "Recommended editing 3rd bubble to "Test miRNA function in AD in vivo models." We have made this edit in the revised manuscript.
Reviewer 2 Report
This review focuses on miRNA as biomarkers of Alzheimer's disease. The article is well presented and the writing is clear.
The purpose or aim of the review is, however, not especially well laid out. Recent meta analyses have summarised the microRNA likely to be altered in Alzheimer's disease and useful as biomarkers (Table 1 is by no means an inclusive summary of Alzheimer's related microRNA).
Section 3 highlights some recent studies and the review then hones in on three microRNA. While I think is is useful to consider particular microRNA in relation to Alzheimer's the work would be improved by justifying the author's focus on these three. The authors might consider how their short list relate to each other.
Further, it would be useful consider if there is value in markers of dementia rather than specific markers of Alzheimer's disease, how a panel of biomarkers might actually be translated into a diagnostic device and how microRNA might be used therapeutically.
Author Response
We thank the reviewer for the opportunity to clarify the objective of our review and address the specific comments as follows.
1. "The purpose or aim of the review is, however, not especially well laid out. Recent meta analyses have summarised the microRNA likely to be altered in Alzheimer's disease and useful as biomarkers (Table 1 is by no means an inclusive summary of Alzheimer's related microRNA)." In the revised manuscript we added the following highlighted text in l.66-70 "The purpose of this review is to highlight evidence that evaluating circulating levels of a set of small ncRNAs might serve as AD prognostic and diagnostic biomarkers for consideration by NIAA-AA research framework [9].We deem this consideration to be particularly timely because RNA therapeutics are emerging as a promising approach in central nervous system (CNS) diseases [33-36]."Our Table 1 features micoRNAs which together with miR-29,-34, and -146 appear among a deregulated set of microRNAs in biospecimens of a larger number of human subjects.
2. "Section 3 highlights some recent studies and the review then hones in on three microRNA. While I think is is useful to consider particular microRNA in relation to Alzheimer's the work would be improved by justifying the author's focus on these three." We thank reviewer for this comment. New highlighted text is added in the revised manuscript in l. 149-152: "Here we review the evidence from multiple studies in support of the notion that a few miRNAs may serve as prognostic and diagnostic AD biomarkers with a mechanistic dimension. This group of candidate AD biomarker miRNAs highlights dysregulation in cellular functions underlying pathophysiologic processes likely years before the manifestation of AD symptoms."
3. "Further, it would be useful consider if there is value in markers of dementia rather than specific markers of Alzheimer's disease, how a panel of biomarkers might actually be translated into a diagnostic device and how microRNA might be used therapeutically." Please see the newly added highlighted text in l. 66-70 of the revised manuscript.
Round 2
Reviewer 1 Report
I am satisfied with the revisions made by the authors and I think it would be acceptable for publication.
Reviewer 2 Report
The manuscript has been improved in the review process, however, it is not comprehensive and the justification for the focus on the microRNA chosen by the authors is remains weak.